# Improving foot self-care in people with diabetes in Ghana: A development and feasibility randomised trial of a context appropriate, family-orientated diabetic footcare intervention

Joseph Ngmenesegre Suglo[1,2]*, Kirsty Winkley[1], Jackie Sturt[1]

1 Florence Nightingale Faculty of Nursing, Midwifery and Palliative Care, King's College London, London, United Kingdom, 2 Department of Nursing and Midwifery, Presbyterian University, Agogo, Ghana

☉ These authors contributed equally to this work.
* suglojoseph@gmail.com

**Data Availability Statement:** All relevant data are within the manuscript and its Supporting Information files.

## Abstract

### Objective

Africa presents a higher diabetic foot ulcer prevalence estimate of 7.2% against global figures of 6.3%. Engaging family members in self-care education interventions has been shown to be effective at preventing diabetes-related foot ulcers. This study culturally adapted and tested the feasibility and acceptability of an evidence-based footcare family intervention in Ghana.

### Methods

The initial phase of the study involved stakeholder engagement, comprising Patient Public Involvement activities and interviews with key informant nurses and people with diabetes (N = 15). In the second phase, adults at risk of diabetes-related foot ulcers and nominated caregivers (N = 50 dyads) participated in an individually randomised feasibility trial of the adapted intervention (N = 25) compared to usual care (N = 25). The study aimed to assess feasibility outcomes and to identify efficacy signals on clinical outcomes at 12 weeks post randomisation. Patient reported outcomes were foot care behaviour, foot self-care efficacy, diabetes knowledge and caregiver diabetes distress.

### Results

Adjustments were made to the evidence-based intervention to reflect the literacy, information needs and preferences of stakeholders and to develop a context appropriate diabetic foot self-care intervention. A feasibility trial was then conducted which met all recruitment, retention, data quality and randomisation progression criteria. At 12 weeks post randomisation, efficacy signals favoured the intervention group on improved footcare behaviour, foot self-care efficacy, diabetes knowledge and reduced diabetes distress. Future

**Funding:** This study is a part of JNS PhD thesis at King's College London funded by the Centre for Doctoral Studies under the Africa International Postgraduate Research Scholarship. The funders had no role in study design, data collection and analysis, decision to publish, or preparation of the manuscript. No grant numbers applicable to scholarship received by JNS.

**Competing interests:** The authors have declared that no competing interests exist.

implementation issues to consider include the staff resources needed to deliver the intervention, family members availability to attend in-person sessions and consideration of remote intervention delivery.

## Conclusion

A contextual family-oriented foot self-care education intervention is feasible, acceptable, and may improve knowledge and self-care with the potential to decrease diabetes-related complications. The education intervention is a strategic approach to improving diabetes care and prevention of foot disease, especially in settings with limited diabetes care resources. Future research will investigate the possibility of remote delivery to better meet patient and staff needs.

## Trial registration

Pan African Clinical Trials Registry (PACTR) - PACTR202201708421484: https://pactr.samrc.ac.za/TrialDisplay.aspx?TrialID=19363 or pactr.samrc.ac.za/Search.aspx.

## Introduction

Diabetes is a public health challenge, with a global estimate of 578 million and 700 million people with the disease by 2030 and 2045, respectively [1]. The most common and expensive complications to treat are diabetic foot ulcers (DFU) [2,3]. People with diabetes have approximately a 25% risk of developing open wounds and neuropathies in their lifetime [4]. This has made DFU the leading cause of non-traumatic amputations [5], with morbidity and mortality related to DFU at almost 50% over a five-year period [6]. This puts economic pressure and stress on health care systems, especially in low-middle-income countries (LMIC) with a higher DFU burden than the global prevalence estimates of 6.3% [7].

High-income countries have significantly reduced DFU burden over the past two decades through patient education, specialist care, clear referral pathways, and multidisciplinary or professional teams [8–10]. These measures are affirmed by international guidance [11,12]. Current research suggests that family members and friends can significantly influence an individual's ability to manage DFU at home [13–16]. Therefore, the engagement of families in diabetes self-management programs may enable families to fulfil a vital role in diabetes care and prevention [17,18].

In populations with strong family ties, interventions that engage both the person with diabetes and family caregivers can enhance self-care behaviours related to medication, diet, coping and problem-solving skills [13,19–21]. In Asia, the Americas and Indonesia, caregivers' involvement in foot inspection, checking of feet sensation, diet/meal planning, and setting of diabetes self-management goals resulted in better diabetes outcomes [21]. Similarly, in most African countries, there is strong kinship and ties with family members who often provide physical, mental, and economic support to people with chronic conditions [22–24].

In Ghana, complications related to diabetes are one of the top ten causes of medically related deaths [25,26]. Despite local efforts in diabetes care, patients and their social networks lack knowledge of evidence-based approaches to prevent or manage the condition [27]. Nonetheless, families and friends in Ghana view caring for an unwell relative as a communal and moral responsibility [22]. Indeed, increasing family engagement is being explored as an

innovative strategy to advance the agenda of the Ministry of Health Ghana (MOH) of enhancing care for people with diabetes and other non-communicable diseases [28].

Given this, there is an imperative to contextualise and evaluate the effectiveness of family-based interventions for the prevention of DFUs in Ghana. A previous systematic review [21] identified a candidate programme for a family-orientated diabetic footcare intervention that might be suitable for delivery in Ghana. The intervention, which had previously been delivered in a LMIC but not evaluated anywhere in Africa, offered self- and family management education and skills training. It demonstrated significant improvement in foot care behaviours, diabetes knowledge, wound healing and DFU incidence [29,30]. This study therefore aimed to:

1. Culturally adapt a family-oriented diabetic foot self-care intervention for families in Ghana informed by stakeholder views.

2. Explore the acceptability of the adapted intervention with people at high risk of DFU and their families.

3. Determine the feasibility of conducting a randomised trial to evaluate the effectiveness of the intervention.

## Materials and methods

### Design

The study comprised two phases (1 and 2):

Phase 1 Intervention development/Adaptation.

The ADAPT guidance for culturally adapting complex interventions to new contexts informed the process [31]. In consultation with the diabetes clinical research lead for the study, we approached people with diabetes, carers, and healthcare professionals in Ghana to join a patient public involvement (PPI) group. This group helped to facilitate the adaptation of an evidence-based intervention identified from our earlier systematic review [21]. The PPI group also provided their opinions on the suitability of the intervention and areas that needed modification. The evidence-based intervention was a "3-month self- and family management support program" which provided participant education on diabetes, diabetic foot and/or wound care skills and encouraged family involvement in diabetic foot ulcer care. Details of all components of this intervention have been published elsewhere [21,32].

Key informants (comprising nurses, people with diabetes, and family caregivers) were then recruited with ethical approval and interviewed to provide further feedback on the intervention with regards to content, delivery, relevance and acceptability. The original diabetic footcare (DFC) intervention was delivered by the researcher (JNS) to key informants, face-to-face in a group setting, before semi-structured interviews commenced. A copy of the interview schedule can be found in S4 File. All key informants were interviewed individually, and all interviews lasted between 35 to 45 minutes. At the conclusion of this phase, the development of the adapted DFC intervention was completed.

Upon the researcher's request, three key informant nurses from phase 1 volunteered to deliver the intervention in phase 2 to patients and their families. All three nurse providers had at least a Bachelor of Science degree in nursing and received an additional four hours of facilitation training provided by the researcher (JNS) before delivering the intervention,

Phase 2: Feasibility trial

A two-arm feasibility randomised trial, with participants equally randomised to either a control or an intervention arm, was conducted to determine the acceptability of the intervention and the feasibility of conducting a future effectiveness evaluation. Standard care at the

clinic comprised regular appointments with a physician and monitoring of blood pressure, weight and blood glucose. Participants in the control group and intervention group received this care as usual. In addition, the intervention group also received the experimental family-oriented intervention.

**Family-orientated foot care intervention.** This was a 2-hour weekly program delivered over four weeks in groups, face-to-face, by trained nurses. The program incorporated education on diabetes and experiential skills workshops on foot checks and self-care. Thematic areas of the program comprised: diabetes and its complications; individual risk factors, diabetes-related foot problems; family roles in diabetic foot ulcer prevention; foot checks/care; foot hygiene; footwear and socks selection; and foot sensitivity checking and physical activity. It also involved providing participants in the intervention group with (i) nail clippers to support training on the proper cutting of toenails (ii) a plastic handy mirror for inspection of the bottom of their feet, (iii) a 10g monofilament for an experiential demonstration on foot sensitivity checking (iv) a bag to contain all equipment. Following self-management education, skills demonstration by the nurse and a return demonstration of foot care by people with diabetes and caregivers, each participant, assisted by their caregiver, was asked to perform daily foot care at home. Full details of the intervention curriculum are presented as a (S5 File).

## Phase 1 and 2 participants and sample size

Study participants in both phases 1 and 2 were people with diabetes and their respective family caregivers who were recruited as a dyad. Professionals (nurses) also participated in phase 1. Consideration was given to the availability and willingness of people with diabetes, carers and healthcare providers [31,33]. Thus, through a purposive sampling technique, 5 family-dyads and 5 nurses were recruited in phase 1 as key informants to guide the development of the intervention.

Feasibility trials do not need a power-based sample size calculation, although a sample with at least 25 dyads per trial arm is deemed a good practice [34–36]. For the purposes of providing a standard deviation (SD) that facilitates a sample size calculation for a future trial, Julious [37] proposes the recruitment of at least n = 12 per group. This is equivalent to N = 24 for a traditional two arm trial which is also similar to sample sizes proposed by other authors [38]. It has been suggested that sample sizes that are less than 70–100 per group will produce large confidence intervals, and if examining the confidence intervals of feasibility process outcomes, then a sample size of at least 70 is recommended [39]. However, these feasibility sample size recommendations should also be based on pragmatic considerations including resource and budgetary constraints. A recent pilot trial of a different intervention in a similar population of people with diabetes in Ghana recruited 26 participants per trial arm for an effect size of 0.8, at a power of 80%, and an alpha level of 5% [40]. In phase 2, fifty dyads were recruited into this feasibility trial. Participants for both phase 1 as key informants and the phase 2 feasibility trial met the eligibility criteria described in Table 1.

## Recruitment and consent process

Recruitment was conducted by the researcher (JNS) and a trained research nurse. All recruitment activities were conducted between February 15 and 30 August 2022 at the diabetes clinic of Konfo Anokye Teaching Hospital (KATH). The diabetes clinic is part of a bigger 1200-bed capacity tertiary hospital centrally located in Kumasi, the administrative capital of the Ashanti region of Ghana. The central location and good skill mix of healthcare professionals means that the clinic receives referred patients from most parts of the country.

**Table 1. Participant eligibility criteria.**

| Eligibility criteria | Phase 1 | Phase 2 |
|---|---|---|
| **Inclusion** | | |
| People with diabetes were included if they are as follows | | |
| Adults aged ≥18 years with a confirmed diagnosis of type 1 or 2 diabetes | √ | √ |
| Fulfil at least one of the following three criteria for definition of high risk for foot ulcers; (i) medically confirmed diagnosis of neuropathy (ii) previous/healed diabetic foot ulcer or (iii) foot abnormalities at risk of ulcer in the opinion of the investigator (iv) venous insufficiency (skin colour change or temperature difference). | √ | √ |
| Have a family caregiver who live with them and/or assisted them in their day-to-day self-care activities and willing to participate in the study with them | √ | √ |
| Resident of Kumasi | √ | √ |
| Able to give informed consent | √ | √ |
| **Family Caregivers were included if they are as follows:** | | |
| People ≥18 years and living with the individual with diabetes and/or assisted them on daily basis with care activities. | √ | √ |
| Willing to participate in the study and willing to attend appointments with their relative that has diabetes for research related activities | √ | √ |
| **Healthcare professionals were included if they are as follows** | | |
| Nurses that are registered with the Nurses and Midwifery Council of Ghana and currently working with people with diabetes | √ | |
| At least two years working experience with people with diabetes | √ | |
| **Exclusion criteria** | | |
| **Exclusion criteria for People with diabetes** | | |
| People with peripheral vascular disease requiring immediate revascularization. | √ | √ |
| People with current foot ulcers since this study is aiming at prevention of ulcers and not management. | √ | √ |
| **Exclusion Criteria for family caregivers** | | |
| Caregivers who are sick and requiring immediate medical attention | √ | √ |
| Caregivers under 18 years of age | √ | √ |
| **Exclusion criteria for healthcare professionals** | | |
| Other cadre of health workers other than registered nurses | √ | |

Potential participants were invited to the study through information posters displayed on key notice boards within the diabetes clinic. The posters were titled "Diabetic footcare education and training in Ghana" and provided the researcher's contact details. Some of these posters were also displayed on key staff areas such as the consulting physicians' table and the nurses' assessment table. A trained research nurse for this study also made a daily verbal announcement every morning before the start of patient consultations when all service users usually arrive and wait for a clinic consultation. Potential participants upon hearing the announcements or seeing the study poster could contact the researcher directly by phone, email or face-to-face. The researcher (JNS) or a trained research nurse was present at the clinic on a daily basis to manage recruitment. People with diabetes and family caregivers were recruited together but asked to contact the researcher separately to express interest and confirm participation. Potential participants were given the information sheet and screened for eligibility by the researcher (JNS).

Eligible participants voluntarily provided individual written informed consent either at the recruitment centre or completed the consent form at home and returned it to the researcher through a delivery service at the researcher's cost. The trial was pre-registered with the Pan African Clinical Trial Registry (PACTR202201708421484) and ethical permission was granted

by King's College London (HR/DP-21/22-26606) and the KATH Institutional Review Board (IRB/AP/139/21).

## Randomisation and blinding

Upon signing the informed consent, participants completed their baseline assessments and measures before randomisation. Randomisation was done at the individual level. Eligible dyad participants were randomly assigned in a 1:1 ratio using a remote computer-generated random number sequence by an independent statistician who was not part of the research team. The independent statistician generated the randomisation sequence prior to the start of all recruitment activities and had no patient contact. A trial coordinator, blinded until this point, accessed the randomisation database and assigned participating dyads to the intervention and control groups. Due to the physical nature of the intervention, participants, the researcher supervising intervention delivery, and providers delivering the intervention were not blinded to allocation groups. However, the outcomes assessor collecting follow up data from patients was unaware of their assigned group. Also, healthcare workers who provided usual care at the diabetes clinic were blind to the assigned groups of participants. Following allocation, participants were advised not to disclose their group to the staff at the diabetes clinic.

## Phase 2 outcomes

Outcomes included feasibility criteria and patient reported outcome measures (PROMS) which were taken at baseline before randomisation and 12 weeks post randomisation. Thus, all baseline data were collected during the period of recruitment (February to August 2022) and follow up data from 1st to 20th December 2022.

**Feasibility measures.** Based on existing evidence from previous studies of self-care interventions, [40] feasibility progressing criteria to a definitive RCT were predefined by the research team. Feasibility measures comprised: 1) evaluations of recruitment; 2) data completion; 3) intervention session attendance; and 4) participant retention. Recruitment rate was calculated as the number who consented to enter the study over the number who were screened as eligible. Research retention rate referred to the proportion of these participants who were available for data collection at 12 weeks follow-up. Similarly, data completion was defined as the percentage of returned and complete questionnaires at 12 weeks. An additional measure included after the study had started was to determine the proportion of people expressing interest in the study who were eventually eligible. This was to identify whether we were able to include majority of interested persons and if not why.

**PROMS**: Patient reported outcomes were collected to assess the feasibility of obtaining relevant measures for a full-scale trial. A nurse who was not part of the research team collected questionnaires and demographic data from each participant at the 12-week follow-up.

1. Foot care behaviour was assessed using the adapted Nottingham Assessment of Functional Footcare (NAFF) measure [41]. NAFF has indicated internal consistency (Cronbach's alpha = 0.61) and good test-retest reliability (r = 0.91, p < 0.001). The 26-item scale was adapted by the study stakeholders, deleting nine questions with items such as "do you use hot water bottles in bed?" which were not considered relevant for people living in high temperature climates. Each NAFF item is scored from 0 to 3 according to the frequency of a practice, and then summed to produce an overall score, with scores > 50 suggesting satisfactory or good foot care behaviour [42].

2. Foot self-care efficacy was assessed using the Foot Care Confidence Scale (FCCS) [43] which has good internal consistency (Cronbach's alpha = 0.92) and has good content

validity as confirmed by diabetes educators. Although re-test reliability has not been reported, all twelve scale items loaded on one factor in a factor analysis, suggesting that all items are required to adequately measure foot self-care efficacy [43]. Each item is scored on a scale of 1 (strongly lack confidence) to 5 (strongly confident). The maximum score for a participant is 60, and the higher the score the more confident the individual feels in the performance of the recommended footcare.

3. The Diabetes Knowledge Questionnaire (DKQ-24) [44] was used to assess the diabetic foot knowledge of both participants with diabetes and their family caregivers. The 24-item questionnaire was validated among people with diabetes and their support people in the US and demonstrated good construct validity and internal consistency (Cronbach's alpha = 0.78). Each item has three response options consisting of: 1) "Yes"; 2) "No"; and 3) "I don't know." Items are checked against answers, and then scored as correct (1) or incorrect (0) with "I don't know" scored as incorrect. The correct items are then summed to attain a total score. A higher score indicates a better knowledge of diabetes.

4. Finally, the Diabetes Distress Scale for Spouses and Partners (DDS-SP) [45] was used to assess the level of distress caregivers had related to their relatives' diabetes. Although there is no reported re-test reliability value, the instrument has demonstrated good internal consistency (Cronbach's alpha = 0.95). The 22-item instrument is scored on a scale of 0 to 4, with 0 indicating no distress at all and 4 representing a great deal of distress. A higher score indicates the presence of a greater level of diabetes-related distress. A meaningful cut-off point of two indicates moderate stress, and is equivalent to feeling, on average, "a little" distressed or greater [45].

## Data analysis

**Phase 1 analysis of interviews.** Data from the individual interviews were audio recorded and transcribed verbatim. A thematic content analysis [46] was used to analyse participants' responses. Firstly, the lead researcher (JNS) familiarised themselves with the data by reading the transcript repeatedly. Then they developed codes which were checked by other team members (JS and KW). Codes with similar meanings were grouped to form categories.

A priority of the analysis was to identify contextual challenges to implementing the intervention, and thus the analysis was supported by the Consolidated Framework for Implementation Research (CFIR) [47]. This consists of five key domains (intervention characteristics, inner setting, outer setting, characteristics of individuals, and process) which affect implementation. The use of CFIR was not to guide the generation of codes but instead to infer how codes related to its constructs. Thus, CFIR was used to identify possible enablers and barriers to a future intervention study.

An evidence matrix was developed, combining empirical findings from the interviews with insights from an earlier systematic review [21] on DFU interventions. Using these two sources of data assisted with understanding any conflicting perspectives emerging within the interview data. It also supported decision making on content, duration and delivery of the final adapted intervention.

**Phase 2 feasibility study analysis.** Data was analysed using Statistical Package for Social Sciences (SPSS) version 22. Participants characteristics, recruitment, retention, intervention session attendance and data completion rates were analysed using descriptive statistics, frequencies and percentages as appropriate to answer the predefined trial feasibility criteria (See Table 2. for criteria). An additional analysis of the percentage of people expressing interest in the study who were determined to be eligible was computed.

**Table 2. Predefined feasibility criteria.**

| Feasibility Outcome | Methods | Progression criteria |
|---|---|---|
| Recruitment | Recruitment log, screening log, record of reasons for declining to participate. | ≥50% of eligible people giving consent and randomised. |
| Retention | Recruitment log, record of participants completing follow up questionnaire. | ≤15 dyads (30%) of participants lost to follow- up. ≥70% retention of participants in the study at 12 weeks post-randomisation. |
| Data quality/completeness of outcome data measure | Outcome data questionnaire booklet assessed. | 12 weeks post-randomisation data complete for ≥70% of participants. |
| Intervention session attendance | Therapist session log of patient attendance. | Intervention group participants attend three to four weekly intervention sessions (= 3.0 mean attendance). |

**Statistical analysis plan.** Despite the trial not being designed or powered to show an effect of the intervention, all patient reported outcomes, means and standard deviations were still computed. The mean differences between groups at follow-up were compared using independent sample t tests, and included only participants who completed the treatment originally allocated. Normality of the data was assessed through a visual inspection of histograms, aiming to identify bell-shaped distributions which are characteristic of a normal distribution. This method allowed for the qualitative evaluation of the distributional properties of the variables under consideration. The t-test was performed as a one-sided statistical test with a significance level set at $\alpha = 0.05$. Our desired effect size was 0.8, which corresponds to a large effect according to Cohen's conventions [48]. The analysis of PROMS was conducted to provide only broad efficacy signals of the intervention that might be found in a future effectiveness trial, rather than to determine a definitive result in this feasibility trial. Thus, p-values are not presented, as the study was not powered to estimate the effect of the intervention. An available-case analysis (Pairwise deletion) approach was utilised to manage missing data. By this mechanism, we maximised all available data for each specific analysis [49]. Thus, if an observation had missing values for certain variables, it was excluded from analyses involving those particular variables, but was still included other analyses where it had complete data.

## Findings

### Phase 1 interviews

Fifteen key informants comprising family dyads (n = 5), caregivers (n = 5) and nurses (n = 5) took part in the interviews. Categories emerging from the interviews with nurses, people with diabetes and families are provided in Tables 3 and 4. alongside supporting quotations.

Whilst people with diabetes and their families were generally positive with regards to the intervention's look, content and practical educational approaches (including skills demonstrations) there was less consensus regarding the length of the intervention and the optimal method of delivery (remote versus face-to-face).

The five nurses felt that the intervention was very relevant to people with diabetes, and that it was important to support family members who were caring for aged patients. A positive facilitator to implementation was that nurses, especially those who were not specialists, believed that delivering the intervention increased their abilities and skills to teach good footcare to their clients. Some nurses reported feeling confident with supervision from their line manager when beginning intervention delivery. They also reported that as their confidence grew, they would be more comfortable providing the education as part of their practice. However, they identified barriers to implementation, with the most prominent being their lack of time to provide the longer sessions required to demonstrate footcare skills to patients. They

**Table 3. Key informant feedback (Patients, Caregivers and Nurses) showing common themes and exemplar quotes.**

| Key informants N = 15 | Patients and caregivers | Nurses |
|---|---|---|
| **Content of intervention** | ***Proactive and interesting***: *"the demonstration aspects of the program were so interactive and engaging. It appeared as if we were playing but learning a lot of things I didn't know. . . I would like to come again"* (P4 with diabetes)<br>***Provides relevant and new knowledge***: *"We were taught how to check our feet and wash our feet. I used to wash my feet but not drying between my toes like we were taught yesterday. This is very good to know"* (P3 with diabetes)<br>***Easy to understand***: *"My mother cannot read because she not been to school . . . but because now you are using many photos and hands-on demonstration it is easy to learn"* (P1 caregiver)<br>***Boosts confidence***: *"with the personal practice at the program and observing others do it, I feel so capable and good . . ., I hope when I go home, I will be able to continue"* (P3 with diabetes) | ***Engaging for patients and staff***: *"We have received similar foot checks training in the past, but it was not engaging like this one that we are provided with the foot care items. This is really good information for us and the patients as well'* (Nurse P2)<br>***Easy to teach and understand***: *"With the footcare items provided, it makes the teaching of patients easier and practical . . . if it were to be only talking, the patients easily forget when they get back home'* (Nurse P3) |
| **Delivery issues** | ***Remote delivery preferred***: *"I wish there was another way I could learn these things without missing my own work. Can it be done in the radio or television. . . or make sessions brief that we can finish and still go to market?"* (Caregiver P 3).<br>***Non suitable venue***: *". . . others like me cannot walk on these stairs because of my knee pains. . . . I don't know if you have another place we can do this education program?"* (P3 with diabetes)<br>***Face-to-face and group delivery helpful***: *". . . you see, because we are all in the room, he (intervention provider) answered all my questions . . . and after I observed the other two people performed the foot checks, I was able to do it too"* (P2 with diabetes) | ***Face-to-face delivery possible only on specific days***: *". . . because of the time involved, meeting to provide the training for patients may be possible only on Thursday or Monday . . . these days not always busy at the clinic. Unlike Tuesdays and Wednesday that the clinic is very heavily attended . . ."* (nurse P2)<br>***Increases workload***: *'. . .You see the skills demonstration and return demonstration is very good for them (patients) to learn but it increases our workload . . . and we the nurses are few compared to our workload'* (nurse P4) |
| **Duration and frequency of intervention** | ***Intervention session too long***: *". . .because I need to go to work . . . " "the program very good but the sessions were too long. . . maybe you can make each day very short only 1hour in the morning. . .over a period"* (P2 Caregiver) | ***Intervention time should be reduced***: *". . .a lot of things that needs doing really . . . but not all can be done. We are constraint in so many ways but is worth preserving small time for patient education like this . . . lets work at making the program very brief to be delivered within shortest possible time"* (nurse p4) |
| **Acceptability of intervention** | ***Importance of intervention for health***: *"I like it . . .very good program,.. my mother diet after amputation because of diabetes . . . may be it could have been prevented if programs like this were available"* (P4 Caregiver)<br>*"This my second time participating in research . . . but always after the research they don't continue the programs at the hospital . . . This is very good education that should be continued by the diabetes clinic authorities after this research"* (P 3 with diabetes)<br>***Is a moral responsibility***: *"it is important to know these things at home . . . when I am old, my children will also take care of me . . . so I have to come and learn to be able to assist her now that she old . . . it is my duty"* (P2 caregiver) | ***A needed intervention to prevent foot disease***: *"Diabetic foot disease is really on the increase; we see them in the clinic every day and so this project seeking to address this is a good idea must be given all the attention and support"* (nurse P3)<br>***Provides valuable skills***: *"I am not a diabetes nurse specialist . . . it is from programs like this that we learn . . . is a good idea. . . ."* (Nurse P5) |
| **Caregiver requirement** | ***Excludes people***: *". . . I have a friend that I know she will be very interested in this program, but she lives alone . . . can you consider her to participate? . . . because all her children are grown and working outside Kumasi"* (P1 with diabetes).<br>*". . .My son goes to work in the morning and only comes home in the evening to assist me . . . if only you could teach us those with diabetes, we can also inform our children and those at home what to do. . . . he (referring to his son) is happy to help me but very busy with work"* (P3 with diabetes) | ***Difficulty getting caregivers***: *"'. . .Is it possible to do it for only patients, because even though the carers are very important in this process, some patients really are not able to bring their caregivers'* (nurse P 2). |

also felt that the requirement to teach people with diabetes and caregivers in scheduled face-to-face sessions could be problematic given that so many family members work. They suggested that while the intervention held great promise and needed to be promoted, there were

**Table 4. Contextual facilitators and barriers to the conduct of the study intervention.**

| CFIR domain | Element of CFIR | Facilitator | Verbatim Quote |
|---|---|---|---|
| Intervention Characteristic | Relative advantage | It is easier to teach patients using equipment | '. . .With the footcare items provided, it makes the teaching of patients easier and practical . . . if it were to be only talking, the patients easily forget when they get back home' (Nurse P3) |
| | Design quality and packaging | Involving carers for older people | '. . .. The idea of involving the caregivers is great especially for the aged and those with bad eyesight . . . and because there is skill demonstration everyone who cannot read will still understand . . .' (Nurse P4) |
| Inner setting | Relative priority | The need to curb the incidence of foot disease | " . . .the number of limb amputations we record every year and the bad open wounds we see in this clinic is not just acceptable, authorities need to do something . . . programs like this may help" (Nurse P1) |
| | Compatibility | Nurses have previously been given similar foot checks training. | 'We have received similar foot checks training in the past, but it was not intensive like this one that we are provided with the foot care items. This is really good information for us and the patients as well' (Nurse P2) |
| | Leadership engagement | Available supervisors | " . . .is alright once there is somebody supervising and if there is anything I am not sure, I can ask for clarification. . . so is okay to conduct future studies like this and make the practice part of our routine" (Nurse P3) |
| Characteristics of individuals | Self-efficacy | Increased self-confidence | "I think I felt more confident delivering the training after the first session. . .I was now used to the steps . . . "(Nurse P4) |
| | Other personal attributes | Readiness to learn the intervention | "I am not a diabetes specialist just like most of my colleagues, it is from programs like this that we learn, it may be a good idea if we are taken for specialist training. . . delivering such an intervention wouldn't have been a problem. . . but we will learn it" (nurse P5) |
| | Executing | Supervision from researcher and/or clinic managers | " . . .you see, doing these things are good but we will always need a senior staff to be around like the way our in charge (referring to unit manager) has been doing . . . because you know we are not diabetes specialist . . ." (Nurse p4) |
| **CFIR domain** | **Element of CFIR** | **Barrier** | ***Verbatim Quote*** |
| Intervention Characteristic | Design quality and packaging | Difficulty in getting caregivers to participate | "Most persons will be interested in this intervention program, but they will not be able to get the caregivers to come with them, . . . is good, but these days all carers are working . . . only supporting when they are back home with their sick relatives" (Nurse P3) |
| | Complexity | Intervention sessions are too long | '. . .Because it is research many of us stayed till the end even though it took longer than I expected . . . In actual practice we may not have all this time regularly' (nurse P2) |
| Inner setting | Relative priority | Nurses and intervention providers have multiple competing roles | 'I usually do a lot of different things like supervision of other staff/students and taking care of patients, so sometimes it is difficult to take part in research programs. . .' (Nurse p5) |
| | Available resources | Limited resources especially at the district and subdistrict level | 'Hmmm sometimes is not like you cannot do it ooo but the resources needed will not always be available. Here may even be better than district and sub-district levels are worst' (Nurse P3) |
| Characteristics of individuals | Self-efficacy | Lack of diabetes specialist knowledge | " . . . these things we know them . . . but sometimes you are just afraid to volunteer and teach it . . . because you are not a specialist and the hospital hasn't taken you for any training .." (Nurse P4) |
| Process | Executing | Increase in workload | " . . .Is a good program but sometimes we are only five nurses on duty and have to attend to over 80 patients hmm, so you don't get the time to teach them all these things". |

also some potential implementation barriers to realising its full potential. Table 4. presents details of inferred facilitators and barriers to the intervention.

**Final intervention adaptation.** The intervention incorporated diabetes education and an experiential skills workshop on foot checks and care at home. Specific content included: diabetes and its complications; diabetes-related foot problems; self-and family management of

diabetes; family roles in DFU prevention; foot checks/care; foot hygiene; footwear and socks selection; foot sensitivity checking and physical activity. Whilst participants were divided in their views regarding the optimal mode of intervention delivery (face-to-face or remote), all effective interventions of a similar nature identified in an earlier systematic review [21] had been delivered face-to-face. It was also not possible to develop the intervention further for remote delivery before the Phase 2 feasibility trial due to pragmatic reasons and resource implications. Thus, the final intervention was delivered face-to-face in groups at the diabetes clinic. Details of the adaptations made to the original candidate intervention as a result of phase 1 of the study can be found in Table 5.

## Phase 2 Feasibility trial results

From February to August 2022, 434 people with diabetes expressed interest in the study and were assessed for eligibility. Out of those assessed, 83 met the inclusion criteria resulting in 83 potential dyads. Fifty of these dyads consented and were randomised. The reasons for study ineligibility and loss to follow up are provided in the CONSORT diagram (Fig 1).

**Table 5. Original and modified version of intervention.**

| Original Intervention | Modified Intervention based on PPI activities, Key Informant interviews and Review (Suglo et al. 2022) | | |
|---|---|---|---|
| Intervention components | Context modifications made to the intervention | Intervention strategies | Original intervention components omitted |
| • Self-management: physical activities, medications, diet, foot care (i.e., wound care), and routine blood sugar control.<br>• Family management: strengthening the family supports on problem solving in DFU problems, establishing family roles in DFU care, and effective involvement in DFU care.<br>• Wound size measurement.<br>• Foot assessment/checking.<br>• Additional hands-on workshops and skills exercises in performing foot care. | **Modifications after PPI:**<br>• Use of low-literacy level written materials.<br>• Delivery of intervention in a native Ghanaian language (Twi) by bilingual nurses.<br>• Mandatory participation with family/support person.<br>• Emphasis on hands-on demonstrations (experiential learning).<br>• Provision of foot self-care kits containing locally available materials (nail clippers, foot care cream, towel, and a handy mirror).<br>• Use of visual images and descriptive pictures of foot problems and practices as poster cards.<br>• Inclusion of specific role of caregiver.<br>• Specific family caregiver roles to emphasise checking patient foot sensitivity; inspection of bottom of feet for the aged and those with vision problems; to remind patient of clinic appointments, medications, exercises, to help with meals preparation and self-care goals.<br>• Inclusion of opportunities for socializing and problem-solving during role play or return demonstration of foot check activities.<br>**Modifications after Key informant Interviews**<br>• Intervention sessions conducted early in the morning that permit others to still go to their official workplace.<br>• Intervention sessions on off-peak days of the clinic<br>• Shortened intervention sessions over four weeks.<br>• Flexibility on who qualifies as a family member to participate.<br>• Training of three nurses to deliver the intervention.<br>• Change of intervention venue to a ground floor room to be more accessible for persons with knee and vision problems. | • Guided group discussion on the role of family members.<br>• Multiple and iterative presentations of key points.<br>• Modelling<br>• Skill mastery through demonstration/return demonstration.<br>• Verbal persuasion through relating the significance of foot care.<br>• Vicarious experience through observation of others, praise, and congratulatory words. | • HbA1c measurement.<br>• Wound size measurement. |

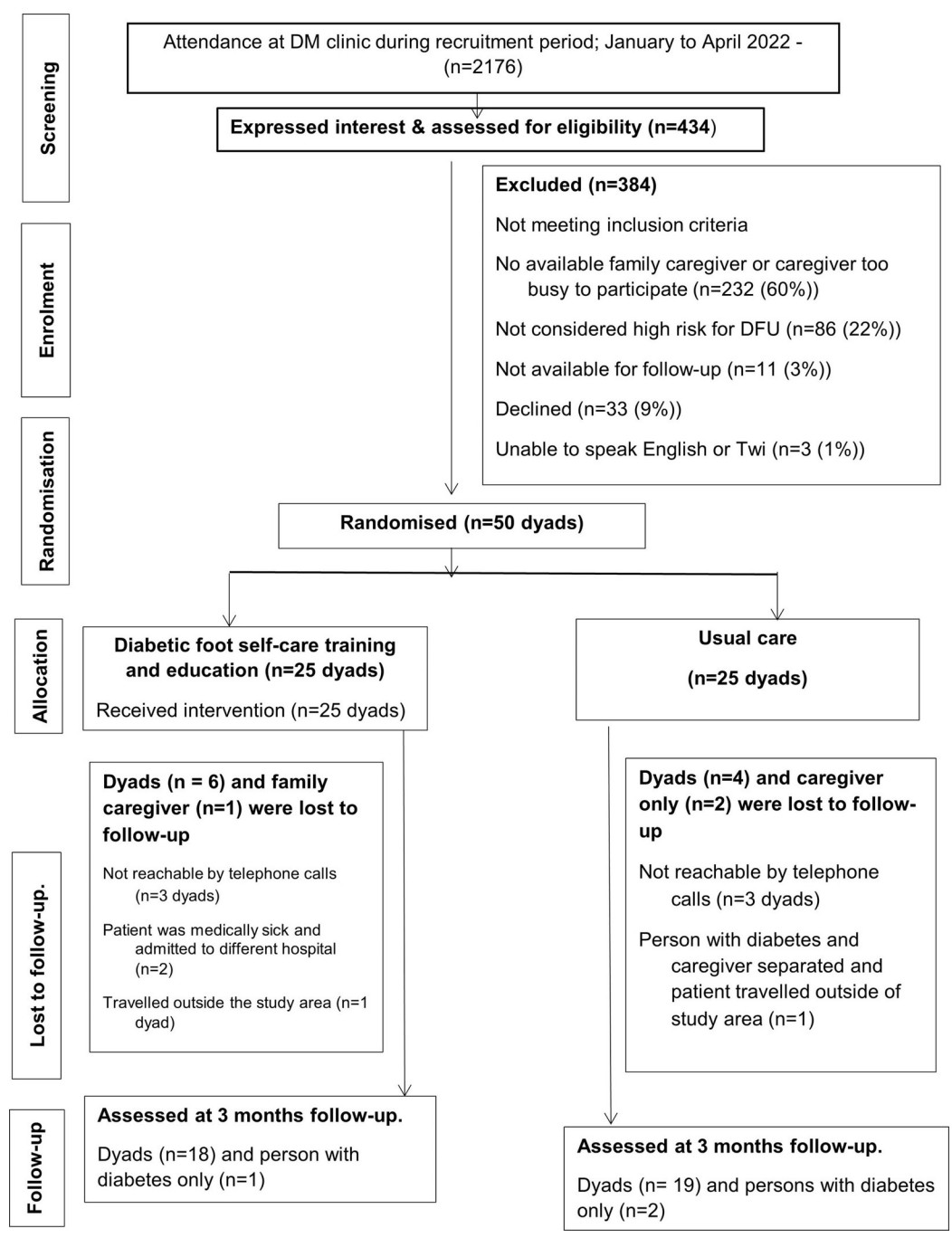

**Fig 1. CONSORT flow diagram.**

The majority of the randomised participants were female with few differences in characteristics between the intervention and usual care groups. In comparison with participants with diabetes, caregivers were younger in age, with a mean age of 44.60 (SD 15.17) and 43.12 (SD 14.15) for the intervention and control groups respectively. Although the majority of the participants in the intervention group were married, most caregivers 27(54%) were sons/

daughters caring for their parents rather than spouses. Demographic details for participants with diabetes and caregivers are presented in Table 6.

**Feasibility outcomes.** All predetermined progression criteria relating to participant recruitment, retention, data completion and attendance were met (see Table 7). Two blood relatives with diabetes were randomised into different groups and there was some sharing of intervention materials between participants resulting in contamination. When determining the feasibility of a trial, it has been recommended that multiple sources of information should be considered and no single feasibility outcome should be considered in isolation [50]. We also computed the proportion of people expressing an interest in the study and who were eligible to participate and reported it as new criterion. In fact, there were a large number of patients who were interested in participating but deemed ineligible for the trial. This was for multiple reasons but in 60% of cases this was because of the requirement that family members also had to be available to take part in the study.

**Patient reported outcomes.** **People with diabetes**: At 12 weeks post-randomisation, participants remaining in the intervention group (n = 19) and control (n = 21) were each analysed in the group they were originally allocated. A between group analysis of NAFF, FCCS and DKQ-24 using independent samples t-tests, all showed that the direction of change favoured the diabetes education program with mean differences of 10.16 on the NAFF (95% C.I. 7.66, 12.67), 8.27 on the FCCS (95% C.I. 5.87, 10.68) and 1.44 on the DKQ-24 (95% C.I. 0.78, 2.09). Although none of the outcome measures provided a minimal clinically important difference, a higher score on each instrument was an indication of improved outcome.

**Family caregivers**: Participants retained in the intervention (n = 18) and control (n = 19) were analysed in their original groups. At follow up, between groups analysis indicated that caregivers in the intervention group had a greater increase in knowledge with a mean difference of 1.34 on the DKQ-24 (95% C.I. 0.84, 1.83) and less diabetes distress with a difference of -12.59 on the DDS-SP (95% C.I. -16.91, -8.27) respectively.

## Discussion and conclusion

### Discussion

The findings from this study suggest that an adapted family orientated DFC intervention was acceptable to people with diabetes in Ghana and their families. Involving stakeholders in the development of healthcare interventions aids successful implementation and can increase patient uptake [51–53]. Engaging with stakeholders from the outset in this study, through PPI and key informants, promoted anchoring and acceptance among both providers and recipients of the DFC intervention. For example, the stakeholder consultation process identified low levels of literacy among participants which led to the development of easier to read text, greater use of foot hygiene images and an emphasis on skills demonstration to support effective learning.

Nurse informants were positive about the purpose and content of the intervention. People with diabetes and their families remained divided on whether face-to-face or remote delivery of the program would be more desirable. Remote or hybrid delivery might be useful to address factors such as urbanisation, migration and job commitments which mean that family members are not always available for face-to-face sessions [54]. Interventions should take advantage of the fast-growing technological advancement in LMIC [55]. A written manual of the intervention with illustrative images may be a useful adjunct to the programme. In addition, videos of the skills demonstration could be circulated via WhatsApp (Meta Platforms) [56] which is popular locally and features content encryption. The WhatsApp platform has been used successfully for facilitated discussions on remote consulting in primary healthcare in a similar

**Table 6. Demographic characteristics of participants.**

| Variables | Diabetic foot education and training (n = 25) | Usual care (n = 25) | Total (N = 50) | P -value* |
|---|---|---|---|---|
| **Demographics of people with diabetes** | | | | |
| Age in years Mean (SD) | 56.48 (10.26) | 55.32 (9.87) | | 0.69 |
| Gender | | | | 0.54 |
| Male | 9 (36.0%) | 6 (24.0) | 15 (30.0%) | |
| Female | 16 (64.0) | 19 (76.0%) | 35 (70.0%) | |
| Marital status | | | | 0.42 |
| Single | 3 (12.0%) | 5 (20%) | 8 (16.0%) | |
| Married | 22 (88.0%) | 19 (76%) | 41 (82.0%) | |
| Others | | 1 (4.0%) | 1 (2.0%) | |
| Duration since diagnosis of diabetes | | | | 0.33 |
| Less than 5 years | 2 (8.0%) | 2 (8.0%) | 4 (8.0%) | |
| 5 to 10 years | 1 (4.0%) | 5 (20.0) | 6 (12.0%) | |
| 11 to 15 years | 8 (32.0%) | 4 (16.0%) | 12 (24.0%) | |
| 16 to 20 years | 10 (40.0%) | 8 (32.0%) | 18 (36.0%) | |
| 21 plus years | 4 (16.0%) | 6 (24.0%) | 10 (20.0%) | |
| Type of diabetes | | | | 0.13 |
| Type 2 diabetes | 22 (88.0%) | 24 (96.0%) | 46 (92.0%) | |
| Type 1 diabetes | 3 (12.0%) | 1 (4.0%) | 4 (8.0%) | |
| Treatment currently being taken. | | | | 0.37 |
| Oral hypoglycaemic agents | 15(60.0%) | 18 (72.0%) | 33 (58.0%) | |
| Insulin and oral hypoglycaemic agents | 10(40.0%) | 7 (28.0%) | 17 (42.0%) | |
| Highest Education | | | | 0.56 |
| No formal education | 5 (20.0%) | 3 (12.0%) | 8 (16.0%) | |
| Basic primary education | 6 (24.0%) | 11 (44.0%) | 17 (34.0%) | |
| Secondary/Tertiary | 14 (56.0%) | 11 (44.0%) | 25 (50.0%) | |
| **Demographic Characteristics of Family caregivers** | | | | |
| Age in years Mean (SD) | 44.60 (15.17) | 43.12 (14.15) | | 0.65 |
| Gender | | | | 0.51 |
| Male | 7 (28%) | 3(22%) | 10 (20.0%) | |
| Female | 18 (72%) | 22(88%) | 40 (80.0%) | |
| Marital status | | | | 0.31 |
| Single | 9 (36%) | 14 (56%) | 23 (46.0%) | |
| Married | 15 (60%) | 11 (44%) | 26 (52.0%) | |
| Other | 1 (4%) | | 1 (4.0%) | |
| Highest Education | | | | 0.63 |
| No formal education | 0 (0%) | 1 (4%) | 1 (2.0%) | |
| Basic primary education | 3 (12%) | 4 (16%) | 7 (14.0%) | |
| Secondary/Tertiary | 22 (88%) | 20 (80%) | 42 (84.0%) | |
| Relationship with the person with diabetes | | | | 0.54 |
| Partner/spouse | 9 (36%) | 5 (20%) | 14 (28.0%) | |
| Father/mother | 11 (44%) | 16 (64%) | 27 (54.0%) | |
| Others | 5 (20%) | 4 (16%) | 9 (18.0%) | |
| Duration been a caregiver in years. | | | | 0,42 |
| Less than 5 years | 3 (12.0%} | 5 (20.0%) | 8 (16.0%) | |
| 5–10 years | 5 (20.0%) | 9 (36.0%) | 14 (28.0%) | |
| Above 10 years | 17 (68.0%) | 11 (44.0%) | 28 (56%) | |

*Statistical significance (p-value < 0.05).

**Table 7. Feasibility outcomes.**

| Outcomes | Protocol criteria | Trial findings | | |
|---|---|---|---|---|
| | | Intervention group (n = 25) | Control group (n = 25) | Trial results (N%) |
| Rate of Recruitment, retention in intervention and research | Identify within 6 months at least 350 dyads express interest in study. | - | - | 434 |
| | 50% of eligible people giving consent and randomised–recruitment rate | - | - | 50 (60%) **Achieved** |
| | ≤15 dyads (30%) of participants lost to follow-up | 6 dyads (24%) | 4 dyads (16%) | 10 dyads (20%). **Achieved** |
| | 70% retention of participants in the study at 12 weeks post- randomisation | 37/50 (74%) | 40/50 (80%) | 77 participants (77%) **Achieved** |
| Quality of outcome data | Baseline data complete for 100% of participants (n = 50 dyads) | - | - | 50 dyads (100%) |
| | 12 weeks post- randomisation data complete for 70% of participants | 34/37 (91%) of returned questionnaire | 36/40 (90%) of returned questionnaire | 70/77 (91%) of returned questionnaire. **Achieved** |
| Intervention session attendance | Intervention group participants attend three to four weekly intervention sessions (3.0 mean attendance). | All experimental participants | - | 3.84 mean attendance **Achieved** |
| Percentage of people expressing interest in the study eligible to participate. | **New criteria** | - | - | 83 (19%) of persons expressing interest in the study |

African setting [57]. Such an approach would ensure that the adaptation of the intervention does not create the health inequalities that digital resources can sometimes cause [31]. Also, a pre-recorded video demonstration of the intervention may boost capacity by saving time for clinic staff who were concerned that understaffing and significant workloads would prohibit effective intervention delivery.

Favourable intervention effects were detected in improving knowledge, footcare behaviour, self-care efficacy and reducing caregiver diabetes distress. Given the feasibility stage of the research and small sample size, p-values were not reported, but the mean difference between groups at follow-up suggest greater improvements in the intervention group. The effect signals have wide confidence intervals due to the small sample size and should be interpreted with caution. However, these indications align with findings on the impact of family interventions for stroke survivors and caregivers [58,59], cancer patients [60,61] and other chronic diseases care [62–64]. They are also supported by previous studies which suggest that family/caregivers provide psychological support, reminders, meal planning and other significant roles that promote people's self-management of their diabetes [13,65–68]. Family engagement has also been shown to reduce the stress associated with caregiving [69]. In addition, nurses reported how the intervention had increased their confidence and ability to support families regarding their foot-care.

Finally, the study confirmed that it is feasible to conduct an effectiveness trial, based on the trial protocol developed, to improve diabetic foot outcomes for people in Ghana. The study met all stipulated progression criteria. However, the protocol can be strengthened in a number of ways. For example, recruitment rates could be increased by allowing people with diabetes who do not attend clinic with their caregiver to participate, with information cascaded to the family by phone, video or through written information. In addition, significant enthusiasm about the program from the outset led to concern regarding being randomised to the control group, with several interested participants declining to participate for this reason. An

alternative study design that permits all participants to receive the intervention at different time points might prove to be more acceptable [70].

Whilst outcome measures in this trial were well completed and reflected expected changes, none were developed and validated within the context of sub-Sahara Africa. Thus, it will be important to ensure their cultural relevance before further trial use. It was not possible to include objective clinical outcomes in this feasibility study. Given the incidence of foot ulcers and amputations in this population, a follow-up of at least 6 to 12 months would have been needed to observe any meaningful changes [71,72]. However, a future effectiveness trial with a longer follow-up duration may be able to assess outcomes such as ulcer rates and wound healing to provide clear information on clinical benefits.

A DFC intervention engaging both family caregivers and people with diabetes in skills training and education is a pragmatic approach. The intervention may confer benefits in terms of better knowledge of foot care, improved self-efficacy and self-management, and a decrease in family distress. These changes are important as they may translate into reduced mortality and morbidity from complications for people with diabetes. Patient education is particularly important in developing countries where foot specialist services and other foot care resources are mostly either not available or not affordable to people with diabetes. Involving informal caregivers generally in the management of chronic conditions is known to be cost-effective and has long-lasting positive effects [73,74].

Cultures with strong family ties may benefit enormously from this family-oriented intervention. Indeed, policymakers could optimize their health expenditure by supporting the involvement of this unpaid caring work by upskilling family caregivers [21]. Recognizing the contextual intricacies and distinct requirements of individuals living with diabetes and their families is paramount. This entails assessing whether the emphasis leans towards "individual" or "family" values [75]. Essential to this understanding is the acknowledgment of the interdependence among family members and their perceptions of health and illness, which are pivotal for successful implementation of family-based interventions. Moreover, ethnic minorities in Western countries, particularly those of African and Asian descent, often exhibit strong familial bonds [76]. Conducting further research and evaluation of family-based diabetes care interventions within these communities could potentially yield significant health benefits.

## Strengths and limitations

This feasibility study was conducted under the same strict conditions of a full-scale trial and remains the first in this sub-region which explores the potential effect and feasibility of family-oriented footcare programs. Although the study had a relatively small sample size and short-term follow-up, it was not intended to be adequately powered to make statistical inferences about effectiveness. Further work on cost-effectiveness will also need to be conducted to determine the feasibility of incorporating it as part of routine practice in Ghana.

## Conclusions

The family-oriented education intervention may have the potential to enhance the footcare behaviour and practice efficacy of people with diabetes through family caregivers' engagement. Findings from this feasibility study have resulted in several recommendations. These include:

1. The study team should explore the potential of adapting some or all of the intervention for remote delivery to fit the work schedules of patients and carers and lessen staff burden.

2. Employing designs such as a waiting list control or a stepped wedge cluster design [70,77] may be an appropriate way of increasing recruitment to the trial.

3. Consideration should be given to the optimal outcome measures for this study and also any adaptations necessary to ensure that measures are culturally appropriate and sufficiently sensitive for studies conducted in sub-Saharan Africa.

## Supporting information

**S1 File. CONSORT checklist.**
(DOC)

**S2 File. Study protocol.**
(DOCX)

**S3 File. Membership of the PPI group.**
(DOCX)

**S4 File. Individual interview guide.**
(DOCX)

**S5 File. Intervention curriculum.**
(DOCX)

**S6 File. Phase 1 Interviews coding framework.**
(DOCX)

**S7 File. SPSS Data set persons with diabetes.**
(SAV)

## Acknowledgments

Thanks to Mrs Agnes Owusu, Dr Philemon Amooba, Dr Frank Botsi Micah, Miss Mavis Mallory Mwinbam, Mr Silas Sebire and the entire Nursing team at the diabetic clinic of Komfo Anokye Teaching Hospital for diverse forms of support during this study. Thanks to Dr Samantha Coster for support in editing and proofreading this paper.

## Author Contributions

**Conceptualization:** Joseph Ngmenesegre Suglo, Kirsty Winkley, Jackie Sturt.

**Data curation:** Joseph Ngmenesegre Suglo.

**Formal analysis:** Joseph Ngmenesegre Suglo.

**Investigation:** Joseph Ngmenesegre Suglo, Kirsty Winkley, Jackie Sturt.

**Methodology:** Joseph Ngmenesegre Suglo, Kirsty Winkley, Jackie Sturt.

**Project administration:** Joseph Ngmenesegre Suglo.

**Resources:** Kirsty Winkley, Jackie Sturt.

**Supervision:** Kirsty Winkley, Jackie Sturt.

**Validation:** Kirsty Winkley, Jackie Sturt.

**Writing – original draft:** Joseph Ngmenesegre Suglo.

**Writing – review & editing:** Kirsty Winkley, Jackie Sturt.

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
