## [Editor Report · Decision Letter 0]

18 Oct 2023

PONE-D-23-33044Improving foot self-care in people with diabetes in Ghana: A development and feasibility study of a context appropriate, family-orientated footcare intervention

PLOS ONE

Dear Dr. Suglo,

Before it can be sent out for peer review, please resubmit the paper together with the correct reporting guideline, i.e. the CONSORT extension for pilot and feasibility trials (https://www.equator-network.org/reporting-guidelines/consort-2010-statement-extension-to-randomised-pilot-and-feasibility-trials).

Please make sure that the fields are adequately addressed in the paper and resemble the filled-in checklist, as in the  currently submitted CONSORT sheet this is not the case. For example, I cannot find any information about items 8-10 (Allocation and Randomisation) although it is stated in the form that these aspects are reported on page 7.

Also, could you provide a direct link for the study registration as with the provided registrartion number, I am unable to find the correct form (which is surely not the authors' fault).

We look forward to receiving your revised manuscript.

Kind regards,

Sascha Köpke

Academic Editor

PLOS ONE

2. Please amend either the title on the online submission form (via Edit Submission) or the title in the manuscript so that they are identical.

---

## [Author Response · Author response to Decision Letter 0]

9 Nov 2023

Dear Professor Sascha Köpke,

Thank you so much for the review and signposting to the resources. That has been very helpful in improving the paper. We have now used the correct reporting guidelines, that is the CONSORT extension for pilot and feasibility trials. We have added some additional text to the paper to adequately reflect the filled checklist. Details of the randomisation process has been added to make it clearer. Also, the sequence of some subheading such as participants, recruitment and randomisation have been reorganised to make the paper easier to follow.

The trial registration link has been added together with the registration number.

We hope you find our manuscript suitable for publication and look forward to hearing from you in due course.

Thank you,

Joseph Suglo

---

## [Decision Letter · Decision Letter 1]

29 Jan 2024

PONE-D-23-33044R1Improving foot self-care in people with diabetes in Ghana: A development and feasibility randomised trial of a context appropriate, family-orientated diabetic footcare interventionPLOS ONE

Dear Dr. Suglo,

Thank you for submitting your manuscript to PLOS ONE. After careful consideration, we feel that it has merit but does not fully meet PLOS ONE’s publication criteria as it currently stands. Therefore, we invite you to submit a revised version of the manuscript that addresses the points raised during the review process.

As you can see, the reviewers have made some suggestions to improve the manuscript. The statistical reviewer (reviewer 1) suggest to present p-values instead or in addition to 95% CIs. I disagree on this aspect and suggest to keep the CIs and only optionally consider adding p-values.

Kind regards,

Sascha Köpke

Academic Editor

PLOS ONE

Journal Requirements:

**Additional Editor Comments:**

The abstracts needs revision, please make sure to transparently report numbers here, i.e. numbers of participants and numbers (together with 95% CIs) for main results, if applicable.

Reviewer's Responses to Questions

**Comments to the Author**

1. If the authors have adequately addressed your comments raised in a previous round of review and you feel that this manuscript is now acceptable for publication, you may indicate that here to bypass the “Comments to the Author” section, enter your conflict of interest statement in the “Confidential to Editor” section, and submit your "Accept" recommendation.

Reviewer #1: (No Response)

Reviewer #2: All comments have been addressed

2. Is the manuscript technically sound, and do the data support the conclusions?

Reviewer #1: Yes

Reviewer #2: Yes

3. Has the statistical analysis been performed appropriately and rigorously? 

Reviewer #1: Yes

Reviewer #2: Yes

4. Have the authors made all data underlying the findings in their manuscript fully available?

Reviewer #1: Yes

Reviewer #2: Yes

5. Is the manuscript presented in an intelligible fashion and written in standard English?

Reviewer #1: Yes

Reviewer #2: Yes

6. Review Comments to the Author

Reviewer #1: This manuscript presents analysis of data generated from a feasibility randomized trial to determine the effectiveness of a diabetic footcare intervention (wrt. a control), in a diabetic population in Ghana. The trial was approved by the respective ethics board, and registered within the PACT registry. The study objectives are on target. However, I mostly have some concerns/comments in the statistical design and analytical framework, which may require attention.

1. Sample size/power: The sample size/power statement provided is not adequate; although this is a feasibility trial, some thoughtful presentation of the anticipated sample size is required. For example, please read: https://www.ncbi.nlm.nih.gov/pmc/articles/PMC8849521/

There should be clear mention of the statistical test used (1-, or 2-sided), the level of significance, and the desired effect size the authors like to see.

Create a separate subsection.

2. Statistical analysis plan: Independent sample t-tests were proposed, which is only valid under assumptions of Normality. Alternative (nonparametric) tests, under violations of Gaussian assumptions, were not mentioned, such as the Wilcoxon rank-sum test. Also, have the authors assessed Gaussian/Normal assumptions of the quantities to be compared via t-tests?

3. Missing Data: How is missing data handled in statistical analysis?

4. Results: At various places, results are stated with estimates, and 95% confidence intervals. While this is nice, adding the p-values would be more relevant.

5. Discussion Section: This section should clearly allude to future studies on other populations and geographical regions to further validate the current findings. The findings from this protocol will only be limited to this population from Ghana.

Reviewer #2: I think the study is well written and all comments raised by previous authors have been adequately addressed.

Abstract: In the abstract, you did not mention the control group the intervention was compared to. It would be appropriate to include it in the abstract.

Introduction: As indicated by the authors, the family-based intervention has already being explored or used by other investigators in LMIC setting and has demonstrated improved outcomes. Has this intervention being used to improve patients outcomes in Ghana?. If yes, how will your study be different from the previously existing studies.

Methods: Line 104: Can you briefly state or describe the evidence-based intervention identified in the previous systematic review?

7. PLOS authors have the option to publish the peer review history of their article (what does this mean?). If published, this will include your full peer review and any attached files.

Reviewer #1: No

Reviewer #2: No

---

## [Author Response · Author response to Decision Letter 1]

19 Feb 2024

Dear Professor Sascha Köpke,

We would like to express our sincere gratitude to you and the reviewers for your thoughtful and constructive feedback on our manuscript entitled "Improving foot self-care in people with diabetes in Ghana: A development and feasibility randomised trial of a context appropriate, family-orientated diabetic footcare intervention”.

We have carefully considered the reviewers' comments and have made revisions to address their concerns. In the attached response to the reviewers file, we provide detailed responses to each of the reviewers' comments.

Thank you once again for your valuable feedback and support.

Joseph Suglo

Florence Nightingale Faculty of Nursing, Midwifery & Palliative Care 

King’s College London 

James Clerk Maxwell Building 1.32 

London

---

## [Decision Letter · Decision Letter 2]

11 Mar 2024

PONE-D-23-33044R2Improving foot self-care in people with diabetes in Ghana: A development and feasibility randomised trial of a context appropriate, family-orientated diabetic footcare interventionPLOS ONE

Dear Dr. Suglo,

Thank you for submitting your manuscript to PLOS ONE. After careful consideration, we feel that it has merit but does not fully meet PLOS ONE’s publication criteria as it currently stands. Therefore, we invite you to submit a revised version of the manuscript that addresses the points raised during the review process.

We look forward to receiving your revised manuscript.

Kind regards,

Sascha Köpke

Academic Editor

PLOS ONE

Journal Requirements:

Additional Editor Comments:

As you have adequaetly revised the manuscript following the reviewers comments, there are only a few editorial comments on minor aspects left that should be addressed in a final revision, mostly concerning the revised text:

    Please check for language and typos throughout, e.g. in the abstract: (a) space missing: “N=50dyads” or (b) article missing: “in [an] individually randomized…” or (c) “greater footcare behavior”, which should rather be “improved footcare behaviour” or similar or e.g. under Design/Pase 1: Details of all components of this intervention is [should be “are” or “have been”] published elsewhere (21,32). These are only examples, so please check the text again.

    Abstract: (a) Please state numbers of dyads for both groups. (b) Please delete “In addition” under “Results”. (c) Although suggested by the reviewer, I would suggest to delete numbers for results here as scales are not described. Alternatively, add scale values and ranges.

    Materials and Methods. Under “PROMS” please report scores and ranges of the presented instruments.

Reviewers' comments:

Reviewer's Responses to Questions

**Comments to the Author**

1. If the authors have adequately addressed your comments raised in a previous round of review and you feel that this manuscript is now acceptable for publication, you may indicate that here to bypass the “Comments to the Author” section, enter your conflict of interest statement in the “Confidential to Editor” section, and submit your "Accept" recommendation.

Reviewer #1: All comments have been addressed

2. Is the manuscript technically sound, and do the data support the conclusions?

Reviewer #1: (No Response)

3. Has the statistical analysis been performed appropriately and rigorously? 

Reviewer #1: (No Response)

4. Have the authors made all data underlying the findings in their manuscript fully available?

Reviewer #1: (No Response)

5. Is the manuscript presented in an intelligible fashion and written in standard English?

Reviewer #1: (No Response)

6. Review Comments to the Author

Reviewer #1: (No Response)

7. PLOS authors have the option to publish the peer review history of their article (what does this mean?). If published, this will include your full peer review and any attached files.

Reviewer #1: No

---

## [Author Response · Author response to Decision Letter 2]

25 Mar 2024

The Academic Editor

PLOS ONE

26th March 2024

Dear Professor Sascha Köpke,

We would like to express our sincere gratitude to you and the reviewers for your thoughtful and constructive feedback on our manuscript entitled "Improving foot self-care in people with diabetes in Ghana: A development and feasibility randomised trial of a context appropriate, family-orientated diabetic footcare intervention”.

We have carefully considered the comments and have revised the manuscript to address all concerns as follows:

Comments: “Please check for language and typos throughout” – 

Response: Thanks so much for the review and the opportunity to proofread our text. We have had the manuscript edited again to make very minor changes throughout to improve grammar and readability.

Comments: Abstract: (a) Please state numbers of dyads for both groups. (b) Please delete “In addition” under “Results”. (c) Although suggested by the reviewer, I would suggest to delete numbers for results here as scales are not described. Alternatively, add scale values and ranges.

Response: The number of dyads for the experimental and control groups have now been stated. We have also deleted numbers for the results as the scales are not described in the abstract. The scale values and ranges are described in the main manuscript.

Comments: “Materials and Methods. Under “PROMS” please report scores and ranges of the presented instruments”.

Response: Thank you for your review. We have now provided this information under patient reported outcome measures section of the manuscript.

Comments: “Please review your reference list to ensure that it is complete and correct” –

Response: We have reviewed our reference list again and have deleted one reference (Amooba 2022). This reference was number 41. It is a PhD student thesis and has now been replaced with a relevant current reference.

Counting on your kind consideration and feedback. Thank you.

Joseph Suglo

Florence Nightingale Faculty of Nursing, Midwifery & Palliative Care 

King’s College London 

James Clerk Maxwell Building 1.32

London.

---

## [Editor Report · Decision Letter 3]

2 Apr 2024

Improving foot self-care in people with diabetes in Ghana: A development and feasibility randomised trial of a context appropriate, family-orientated diabetic footcare intervention

PONE-D-23-33044R3

Dear Dr. Suglo,

We’re pleased to inform you that your manuscript has been judged scientifically suitable for publication and will be formally accepted for publication once it meets all outstanding technical requirements.

Kind regards,

Sascha Köpke

Academic Editor

PLOS ONE